# Correlations between the NMR Lipoprotein Profile, *APOE* Genotype, and Cholesterol Efflux Capacity of Fasting Plasma from Cognitively Healthy Elderly Adults

**DOI:** 10.3390/ijms24032186

**Published:** 2023-01-22

**Authors:** Itziar de Rojas, Laura del Barrio, Isabel Hernández, Laura Montrreal, Pablo García-González, Marta Marquié, Sergi Valero, Amanda Cano, Adelina Orellana, Mercè Boada, Santos Mañes, Agustín Ruiz

**Affiliations:** 1Research Center and Memory Clinic, ACE Alzheimer Center Barcelona, Universitat Internacional de Catalunya (UIC), 08029 Barcelona, Spain; 2Centro de Investigación Biomédica en Red Sobre Enfermedades Neurodegenerativas (CIBERNED), Instituto de Salud Carlos III, 28029 Madrid, Spain; 3Department of Immunology and Oncology, Centro Nacional Biotecnología (CNB-CSIC), 28049 Madrid, Spain

**Keywords:** cholesterol efflux capacity (CEC), *APOE*, Alzheimer’s disease, elderly adults

## Abstract

Cholesterol efflux capacity (CEC) is of interest given its potential relationship with several important clinical conditions including Alzheimer’s disease. The inactivation of the *APOE* locus in mouse models supports the idea that it is involved in determining the CEC. With that in mind, we examine the impact of the plasma metabolome profile and the *APOE* genotype on the CEC in cognitively healthy elderly subjects. The study subjects were 144 unrelated healthy individuals. The plasma CEC was determined by exposing cultured mouse macrophages treated with BODIPY-cholesterol to human plasma. The metabolome profile was determined using NMR techniques. Multiple regression was performed to identify the most important predictors of CEC, as well as the NMR features most strongly associated with the *APOE* genotype. Plasma 3-hydroxybutyrate was the variable most strongly correlated with the CEC (r = 0.365; *p* = 7.3 × 10^−6^). Male sex was associated with a stronger CEC (r = −0.326, *p* = 6.8 × 10^−5^). Most of the NMR particles associated with the CEC did not correlate with the *APOE* genotype. The NMR metabolomics results confirmed the *APOE* genotype to have a huge effect on the concentration of plasma lipoprotein particles as well as those of other molecules including omega-3 fatty acids. In conclusion, the CEC of human plasma was associated with ketone body concentration, sex, and (to a lesser extent) the other features of the plasma lipoprotein profile. The *APOE* genotype exerted only a weak effect on the CEC via the modulation of the lipoprotein profile. The *APOE* locus was associated with omega-3 fatty acid levels independent of the plasma cholesterol level.

## 1. Introduction

All mammalian cells need free (i.e., non-esterified) cholesterol to maintain the structure, fluidity, and permeability of their membranes [1]. However, since most cells outside the liver lack the machinery to metabolize cholesterol, homeostatic mechanisms are required that prevent its harmful accumulation in the extrahepatic periphery. The reverse cholesterol transport (RCT) pathway is involved in transporting cholesterol away from these peripheral tissues back to the liver for excretion into the bile, either as free cholesterol or bile acids. The first step in this pathway is the active transfer of free cholesterol from the peripheral cells to extracellular acceptors, mainly apolipoprotein (Apo) A1-containing high-density lipoproteins (HDLs). This efflux process is mediated by at least four different pathways: simple diffusion, facilitated diffusion mediated by the interaction of HDL with the scavenger receptor B1 (SR-B1), and the ATP-binding cassette transporter subfamily A member-1 (ABCA1) and ATP-binding cassette transporter subfamily G member-1 (ABCG1) in the presence of extracellular acceptors [2]. Following its efflux to HDL, the free cholesterol is then esterified in the plasma and transported to the liver, either directly or after its transfer to apolipoprotein B (ApoB) containing low-density lipoproteins (LDLs) [3]. SR-B1 also participates in the last step of RCT by facilitating the selective delivery of cholesterol-loaded HDL particles to hepatocytes [4].

The cholesterol efflux capacity (CEC), determined via an in vitro assay, reflects the acceptor capacity of biological fluids (a dilution of an Apo B-depleted human plasma sample is commonly used) to extract cholesterol (radio- or fluorescence-labeled) from human or mouse macrophages [5]. This assay can be used to compare the CEC of different individuals. The CEC trait or endophenotype is being investigated for its relationship with important health conditions including polycystic ovarian syndrome [6], acute cardiovascular events [7], metabolic syndrome [8], and Alzheimer’s disease (AD) [9].

Via a pathway analysis based on a combination of large meta-analyses of genome-wide association (GWAS) and genome-wide epistatic (GWES) studies of AD, the International Genomics Of Alzheimer Project (IGAP) highlighted cholesterol transport as strongly influencing the onset and progression of AD [10]. This might be directly related to the pleiotropy of the *APOE* locus. Some *APOE* genotypes are the most important risk factor for late-onset AD [11], but others can affect plasma lipid levels, the appearance of atherosclerosis, cholesterol homeostasis, and other phenotypes [12]. Human APOE protein has three major isoforms depending on the combination of two polymorphisms located at positions 112 (rs429358 [C>T]) and 158 (rs7412 [C>T]). The most common isoform, APOE3, has a cysteine at position 112 and an arginine at position 158. APOE2, the least common isoform, has a cysteine at both positions. The product of the AD risk allele *APOE4* has an arginine at both positions. These amino acid substitutions result in a conformational change that brings together the N-terminal and C-terminal domains of APOE4 (these are normally separated in the APOE2 and APOE3 isoforms). A single *APOE4* allele increases the risk of developing AD by 2–4 times; having two *APOE4* alleles increases this risk by 8- to 12-fold, although the final risk also depends on the genetic background and sex [13].

The inactivation of the *APOE* locus in animal models supports the idea that it is involved in the CEC. The plasma from homozygous *APOE* knockout mice (*APOE*^−/−^) shows a 48–74% reduction in the CEC compared to that of wild-type mice [14]. The overexpression of human *APOE* in macrophages derived from *APOE*^−/−^ mice restores normal CEC values by replenishing APOE-containing particles in the plasma of APOE-deficient mice [14]. Evidence of APOE isoform involvement in the CEC has also been observed in humans [15,16,17,18]. However, large studies have shown that the genetic signals surrounding the *APOE* locus that are associated with the cerebrospinal fluid (CSF) or plasma CEC might be independent of the common *APOE* haplotypes configuring the E2/E3/E4 APOE isoforms [19,20].

The present work examines the relationships between the CEC, the AD-associated SNPs within the *APOE* locus, the demographic features, the clinical phenotypes, and the plasma metabolome in the fasting plasma samples from a cohort of elderly but cognitively healthy patients. While the *APOE* genotype had a pervasive effect on the plasma metabolome, it had only a small nonsignificant effect on the plasma CEC. The plasma metabolite most strongly associated with CEC was 3-hydroxybutyrate, suggesting that ketone bodies (KBs) modulate the CEC of human plasma. Treatment with 3-hydroxybutyrate might therefore help combat several important human conditions in which the CEC is defective.

## 2. Results

### 2.1. Effect of the APOE Genotype on the Plasma Cholesterol Efflux Capacity

After the normalization of plasma CEC values and the exclusion of outliers (Figure 1), a significant difference between *APOE* ε4 carriers and noncarriers was observed in terms of plasma CEC (values for non-carriers were lower) (*n* = 137, Pearson’s r = 0.176, *p* = 0.0397; Figure 1 left panel). However, when stratified by the *APOE* genotype, no significant differences were detected (Figure 1, right panels).

### 2.2. Factors Associated with Plasma CEC

The correlations of different plasma lipoproteins, metabolites and clinical variables between the sex, *APOE* genotype and the plasma CEC were then examined. Figure 2 and Table 1 show the different factors identified as potentially having an influence on the CEC. However, multiple testing corrections and exploratory Pearson’s correlation analysis confirmed only three. The most strongly (and positively) correlated was plasma 3-hydroxybutyrate (mean = 0.127 mmol/L; range = [0.0394–0.439]; r = 0.365; *p* = 7.3 × 10^−6^). Another ketone body (KB), acetoacetate, was similarly correlated with the plasma CEC (Table 1). However, given the strong collinearity among KB concentrations (r = 0.856, *p* = 3.09 × 10^−42^), this latter correlation did not persist when 3-hydroxybutyrate was contemplated as a covariate.

Sex was the variable most strongly associated with the plasma CEC, with male sex showing the stronger influence (r = −0.326, *p* = 6.8 × 10^−5^). Other factors, such as the mean diameter of the LDL particles (LDL-D), a family background of Parkinson’s disease (PD), and plasma albumin, all correlated significantly with the plasma CEC but never reached significance as corrected for the multiple testing procedure (Table 1 and Appendix A). Interestingly, most of the metabolites associated with the CEC did not correlate with the *APOE* genotype, suggesting that APOE isoforms not only have a limited impact on the human plasma CEC, but also on the factors that influence the CEC (Figure 2 and Table 1). However, the absolute content of cholesterol and cholesterol esters transported by small HDL particles (S-HDL-C and S-HDL-CE, respectively) showed a modest positive correlation with both the CEC and *APOE* genotype (r > 0.221, *p* < 0.008; Table 1).

Backward stepwise multiple linear regression, including the top variables, correlated with the CEC (Table 1), *APOE*, and age as potential confounders, identified the most parsimonious model explaining the variance in the CEC. A family background of PD (β = −0.219, *p* = 0.003), sex (β = −0.171, *p* = 0.03), cholesterol esters transported by small HDL particles (S-HDL-CE; β = 0.184, *p* = 0.014), the total-cholesterol-to-total-lipid ratio for very small VLDL particles (XS-VLDL-C-%; β = −0.170, *p* = 0.03), and absolute 3-hydroxybutyrate (β = 0.296, *p* = 0.00018) explained 28.4% of the CEC variance (measured as corrected-r^2^). The final explanatory model contained no non-significant variables, minimizing any potential overfitting (albumin was removed in the eight iteration, age in the seventh iteration, and *APOE* genotype in the sixth iteration).

### 2.3. 3-Hydroxybutyrate Has No Direct Effect on Plasma Cholesterol Efflux Capacity

Since 3-hydroxybutyrate was the metabolite most strongly associated with the CEC, the concentration of this KB was artificially altered to see if there was any direct effect on the CEC. A wide range of concentrations was tested since 3-hydroxybutyrate can rise to 7 mM in fasted healthy people [21]. However, it was found that it did not directly affect the CEC (Figure 3).

### 2.4. Effect of APOE on the Human Plasma NMR Profile

All of the above results suggest that the influence of the *APOE* genotype on the plasma CEC is indirect—if there is any at all. It was therefore hypothesized that the connection between *APOE* and AD risk might be manifested via alternative pathways. To test this idea, associations were sought between the *APOE* genotype and the top plasma NMR variables (as determined by the high-throughput proton nuclear NMR metabolomics technique). Interestingly, many correlations were detected (Figure 2; Appendix A). Among them, 31 NMR variables (13.7%) correlated with the *APOE* genotype but with *p*-values below the corrected significance threshold for multiple testing (*p* < 2.2 × 10^−4^) (Figure 2; Table 2). Importantly, the strongest positive correlations with the ɛ4 allele were observed for the total-cholesterol-to-total-lipid ratio for the large LDL (L-LDL-C-%) and the intermediate density lipoprotein (IDL-C-%) (r > 0.405; *p* < 5.8 × 10^−7^). Conversely, the strongest negative correlations with ε4 were for the total-triglyceride-to-total-lipid ratio for the large LDL (L-LDL-TG-%; r = −0.398, *p* = 9.4 × 10^−7^) and medium-sized LDL particles (M-LDL-TG-%; r = −0.386, *p* = 2.0 × 10^−6^).

These results confirm the huge effect of the *APOE* genotype on the fasting lipid profile of human plasma (Figure 2). To identify the set of independent metabolites firmly associated with *APOE*, a new backward stepwise multiple regression analysis was performed including the 31 variables most strongly correlated with *APOE* (Table 2), together with age, gender, and the presence of dyslipidemia or hypertension. This analysis retained 13 variables after contemplating the covariates (Table 3). Three of these—the total-cholesterol-to-total-lipid ratio for IDL particles (IDL-C-%; β = 5.373, *p* = 0.1.6 × 10^−5^), total esterified cholesterol (EstC; β = −4.59; *p* = 1.7 × 10^−5^), and the total-cholesterol-to-total-lipid ratio for small LDL particles (S-LDL-C-%; β = −7.003, *p* = 1.8 × 10^−4^)—reached the corrected significance threshold. Figure 4 shows the dose-dependent correlation between the top variable and the *APOE* genotype.

Despite the intensive covariation contemplated, the omega-3 fatty acid levels (FAw3) were retained in a univariate backward regression model (*p* = 0.002) (Table 2). This suggests a potential direct effect of the *APOE* genotype on the FAw3 levels independent of all other variables associated with *APOE* (such as the IDL/LDL lipoprotein characteristics, cholesterol/cholesterol ester levels, or clinical covariates). The effect of *APOE* was thus examined as a random factor in the univariate generalized linear analysis, with age, gender, the presence of hypertension or dyslipidemia, and L-HDL-C% (the most strongly correlated of the examined lipids) contemplated as covariates. The association of *APOE* with FAw3 persisted (*p* = 0.001), suggesting an independent effect of *APOE* on plasma omega-3 levels (Figure 4).

## 3. Discussion

The CEC is an important physiological variable associated with certain chronic health conditions [22]. Plasma/serum CEC assays provide a simple means of determining how genetic mutations, medications, or disease states affect the transfer of free cholesterol to serum acceptors—a limiting step in the RCT pathway. Although the CEC was initially conceived as an indirect assay for determining the circulating apolipoproteins (the main serum cholesterol acceptors), it is now known to be a better predictor of cardiovascular risk than HDL levels [7]. One of the strengths of determining the CEC is that it throws light on the activity of the plasma as a whole, including non-apolipoprotein-related factors, in triggering the efflux of cholesterol. Indeed, the use of the CEC has identified circulating plasminogen as a promoter of cholesterol efflux via the ABCA1 pathway [23] and the key role of SR-B1 in this process [4]. In the present work, plasma KBs were found to be potentially critical regulators of the CEC.

The plasma CEC assay used in the present study was based on the BODIPY-cholesterol loading of cAMP-stimulated mouse J774 macrophage-like cells (peripheral donor cells), and apoB-depleted plasma as a cholesterol acceptor. J774 cells were chosen because they do not express APOE [24]. Since the initial aim was to determine the effect of serum *APOE* variants on the CEC, it was believed this cell system would not interfere with any endogenous APOE secretion. It has been suggested that mouse-derived macrophage cell lines mediate the CEC largely by ABCA1-independent mechanisms [25]. Thus, the J774 cells were incubated with cAMP to stimulate ABCA1 expression, therefore favoring ABCA1-mediated cholesterol efflux over aqueous diffusion [3]. This might be important since APOE increases HDL biogenesis in vitro by interacting with ABCA1 [26].

A striking observation in the present work was the strong association between fasted plasma KB and the CEC. However, this plasma was collected from a cognitively healthy (albeit elderly) population; it remains to be seen if this correlation is affected by other neurodegenerative factors circulating in patients with AD. At first glance, these observations seem physiologically meaningful, since KBs are produced by the liver and used in the periphery as an energy source when glucose is not readily available [27], e.g., during fasting or exercise. Bearing in mind this link between KBs and energy metabolism, ketone molecules might contribute to the mobilization of lipids from cellular reservoirs via ABCA1-related cholesterol efflux pathways. However, the mechanism connecting these processes is poorly understood.

While high circulating KB levels correlated with the plasma CEC, their effect would seem to be indirect, since incubating macrophages with 3-hydroxybutyrate did not affect the CEC. It might be that an increase in 3-hydroxybutyrate is associated with a consequent increase in the synthesis and secretion of intestinal APOA-IV-containing lipoproteins [28]. One study has suggested that 3-hydroxybutrate stimulates *APOA-IV* expression. In addition, conditioned media from 3-hydroxybutrate-treated Caco-2 cells can increase the cholesterol efflux by 20–30% [28]. However, this was not observed in the present NMR-derived lipidome data, since the effect of the KB on the CEC was larger than for any co-circulating lipoprotein particle. In addition, the lipoprotein particles most strongly associated with the CEC are independent factors that influence the variance in the CEC. Therefore, although the present data strengthen the link between KBs and the plasma CEC, the former might influence the latter via indirect mechanisms that are independent of increased *APOA-IV* expression. However, the experiments performed here cannot rule out a direct effect of KBs in human cells.

The present results suggest that the KB pathway could be targeted in order to modulate the CEC for therapeutic purposes. This might be especially important in persons with a low plasma CEC capacity (with the associated effect on cardiovascular risk that this entails). The therapeutic potential of KBs is in line with the very recent observation that 3-hydroxybutrate has an anti-arteriosclerotic effect in *APOE* KO mice fed a high-fat diet [29,30]. In the latter work, 3-hydroxybutrate treatment not only inhibited HFD-induced atherosclerosis but reduced hepatic steatosis without affecting body weight gain [29]. These activities of 3-hydroxybutrate were associated with the regulation of hepatic genes involved in lipid/glucose metabolism. Further, ABCA1 is strongly induced by 3-hydroxybutrate [29]. As previously described, ABCA1 is a plasma membrane transporter that mediates the efflux of free cholesterol and phospholipids to lipid-poor apolipoprotein A-I (APOA-I), and it is critical in the biogenesis of HDL particles involved in the early steps of the RCT pathway.

Large meta-analyses of GWAS [10] and other studies [9,31] have suggested that the CEC is casually linked with AD. The role of cholesterol metabolism in AD risk and pathogenesis is also supported by cellular and mouse models as well as by epidemiological data (reviewed in [32]). Certain observations on the role of APOE lipidation in mouse models of β-amyloid (Aβ) deposition in AD brains support the cholesterol hypothesis of AD. Indeed, Aβ formation in cellular models is hampered by cholesterol depletion, cholesterol biosynthesis repression, and acyl-coenzyme A cholesterol acyltransferase (ACAT) inhibition [33,34,35]. In addition, amyloid accumulation in the mouse models of AD is exacerbated by diet-induced hypercholesterolemia [36] and reduced by statin treatment [36,37]. In contrast, other authors have recently reported *APOE* effects on amyloid accumulation to be strictly independent of the CEC [38]. The present results agree with this latter idea, since the *APOE* AD-related genotypes, which are undisputedly associated with AD, had little (if any) effect on the plasma CEC. Indeed, the weak effects observed in the *APOE* e4 carriers would seem to be indirect, most probably via changes in the level of certain plasma lipids and lipoprotein particles associated with the CEC. This conclusion should, however, be taken with caution given the small sample size tested and the inherent uncertainties associated with the variability of human plasma variables. In fact, a large GWAS study showed the association between the CEC and some genotypes close to the *APOE* transcript [20]. However, the observed effect size for CEC variants associated with J774-stimulated CEC was relatively small (beta = −0.193, less than 0.2SDV of CEC per rs445925 allele). Importantly, the reported sentinel variant (rs445925) is not associated with Alzheimer’s disease. In contrast, this variant is highly associated with several lipid metabolites in different studies (see GWAS catalog rs445925 entry for further details). These observations reinforce the notion that SNPs around the *APOE* gene are linked to the CEC via the alteration of the lipid profile of the acceptor. The lack of association to AD of rs445925 is also making it even more difficult to detect the *APOE* effect on the CEC using our experimental design. We think we are not powered enough to reveal such an effect using the current sample size and the selected SNPs (AD-related *APOE* SNPs).

Large Mendelian randomization studies might help to clarify the role of the CEC pathway in the amyloid metabolism, APOE pathophysiology, and the risk of AD. It may be, however, that the connection between the CEC and APOE, particularly its relationship with AD, is restricted to the central nervous system. If this is the case, analyses would necessarily involve the use of specific CNS cells and/or cerebrospinal fluid. Interestingly, KBs (the plasma metabolites most strongly associated with the CEC) have also been linked to AD pathophysiology, and their supplementation has been proposed as a potential treatment for AD [39].

Our study has also limitations. First, we used a relatively low number of subjects. Second, we only applied a single experimental condition for measuring the CEC of human plasma. Third, we only focused on cognitively aged healthy populations. So, it is possible that our observations cannot be generalized to all age ranges. Further studies are necessary to confirm these results.

In conclusion, no direct relationship was found between the *APOE* genotype and plasma CEC in aged individuals. However, a strong association was seen between the latter variable and plasma 3-hydroxybutrate, which is deserving of further investigation.

## 4. Methods

### 4.1. Patients and Samples

The study subjects were a cohort (an ecological sample) of 144 individuals attending the ACE Alzheimer Center Barcelona Memory Clinic (Barcelona, Spain) (mean age: 65.6 ± 8.8 years; 56.9% females). These subjects only showed signs of subjective cognitive decline (SCD) or none at all, and were endorsed as healthy controls [40]. Cognitive assessment was performed as described elsewhere [41]. All subjects had a Mini-Mental State Examination (MMSE) score of ≥27 and a Clinical Dementia Rating (CDR) of 0. Their performance (adjusted for age and educational level) with the *Fundació ACE* Neuropsychological Battery (NBACE) [42] was within the normal range. Table 4 summarizes other demographic information.

The blood samples and *APOE* AD-related genotypes were determined as previously described [40]. Briefly, blood samples were collected, following an overnight fast, in polypropylene vials containing EDTA and immediately refrigerated. Plasma samples were obtained from fresh blood samples drawn by venipuncture into Becton Dickinson Vacutainer tubes (K2-EDTA) using conventional procedures and centrifuged within 2 h. Supernatants containing the plasma fraction were aliquoted and stored at −80 °C until use. Biochemical and hematological measurements were determined at a reference laboratory according to routine clinical standards.

### 4.2. Determination of the Cholesterol Efflux Capacity

The macrophage-specific CEC was determined using BODIPY-cholesterol (MAK192) (Sigma-Aldrich, St. Louis, MO, USA) according to the manufacturer’s instructions. Briefly, J774.1 macrophages (1 × 10^5^ cells/well) were plated into a 96-well plates containing RPMI-1640 medium (BioWest, Nuaillé, France) with 10% fetal bovine serum (FBS) (Capricorn Scientific, Hessen, Germany), 2 mM glutamine, 1 mM sodium pyruvate, and antibiotics. This mixture was then allowed to adhere for 2 h at 37 °C. After washing with serum-free medium, cells were incubated for 16 h at 37 °C with a reaction mixture containing equilibration buffer, acyl-CoA: cholesterol acyltransferase (ACAT) inhibitor (2 μg/mL; Sigma-Aldrich), cyclic adenosine monophosphate (cAMP, 2 μg/mL; Sigma-Aldrich), and BODIPY-labeled cholesterol. Apolipoprotein B-depleted plasma was prepared by PEG precipitation 13% (*v*/*v*) in 200 mM glycine buffer pH 7.4. ApoB-containing lipoproteins from plasma were precipitated by centrifugation (8000 rcf, 20 min 4 °C), and the supernatant was collected and used as a cholesterol acceptor by incubation (4 h, 37 °C) with the BODIPY-cholesterol-labeled cells. The BODIPY-cholesterol fluorescence was determined in the supernatant and in cell extracts using a microplate reader (FilterMax F5, Molecular Devices) (excitement: 485 nm, emission: 595 nm). The cell monolayer was then solubilized with the buffer supplied in the cholesterol efflux assay kit (30 min in the dark on a shaker). The CEC was calculated as the fluorescence counts in the acceptor medium divided by the sum of fluorescence counts in the medium plus the cell lysate. Background cholesterol efflux, in the absence of ApoB-depleted plasma, was subtracted from individual cholesterol efflux values. To correct for inter-assay variability, the results were normalized against the measured CEC for a pooled reference apolipoprotein B-depleted plasma in each assay. All the samples were processed in triplicate, and the mean value was recorded. The intra-assay coefficient of variation (CV) was 7.08% and the inter-assay CV 19.8%, calculated from four CEC values using the pooled plasma as the acceptor.

To study the influence of 3-hydroxybutyrate on the CEC, the reference pooled Apo B-depleted plasma was supplemented with (±)-sodium 3-hydroxybutyrate (Sigma-Aldrich) at different concentrations, and the CEC was determined as above.

### 4.3. High-Throughput Proton Nuclear Magnetic Resonance Metabolomics Profiling

The metabolic markers were quantified in fasted EDTA-plasma samples using a previously described high-throughput proton nuclear magnetic resonance (NMR) metabolomics technique [43,44], performed by Nightingale Ltd. (Helsinki, Finland). The method followed provides a detailed plasma lipid profile, quantifying the routine lipids, lipoprotein subclass, 14 lipid subclasses, fatty acid compositions, and a number of low-molecular-weight metabolites (including amino acids, KBs, and gluconeogenesis-related metabolites).

### 4.4. Statistical Analysis

All data were examined for normality, skew, outliers, and range restrictions. The CEC values were log-normalized. Outliers were identified separately in *APOE4* carriers and non-carriers groups using the R base function ‘boxplot.stats’ with standard parameters (coef = 1.5). Then, 7 individuals from the *APOE4* non-carriers were removed from further analysis. The Pearson correlation analyses were conducted to explore the entire dataset (i.e., demographics, comorbidities, plasma metabolites, CEC, and *APOE* genotype), contemplating the effect in terms of the dose that the *APOE* alleles carried by each subject (i.e., [E2 = −1; E3 = 0; E4 = 1]) [45].

To avoid collinearity problems between the CEC predictors and/or overfitting of the resulting model, the most important potential predictors were selected by backward regression and unsupervised random decision forest analysis [46]. This strategy provides a highly discriminative result, comparable to that obtainable using a classical tree decision technique which avoids overfitting. Ten thousand trees were executed including all predictors for the CEC. The gain ratio rule was used as a splitting criterion [47]. The Knime Analytics Platform v.350 was used for the random forest decision analysis. There are no standard criteria for deciding how many predictors should be selected for later standard inferential analysis. In the present work, the top predictors were selected to explain the CEC phenotype. These predictors were later analyzed together in multivariate linear regression, using a stepwise selection procedure to identify the significant variables. A test for multicollinearity was performed within each regression analysis. All multiple regression analyses were performed by contemplating the CEC/*APOE* dosage as a continuous variable. Correction for the multiple testing procedure was achieved using the Bonferroni correction (246 comparisons); the corrected significance threshold was set at *p* < 2.2 × 10^−4^. All the hypotheses were two-tail-tested at the 95% confidence level. All the calculations were performed using SPSS (version 20.0; SPSS Inc., Chicago, IL, USA) and R (version 1.3.1073).

## Figures and Tables

**Figure 1 ijms-24-02186-f001:**
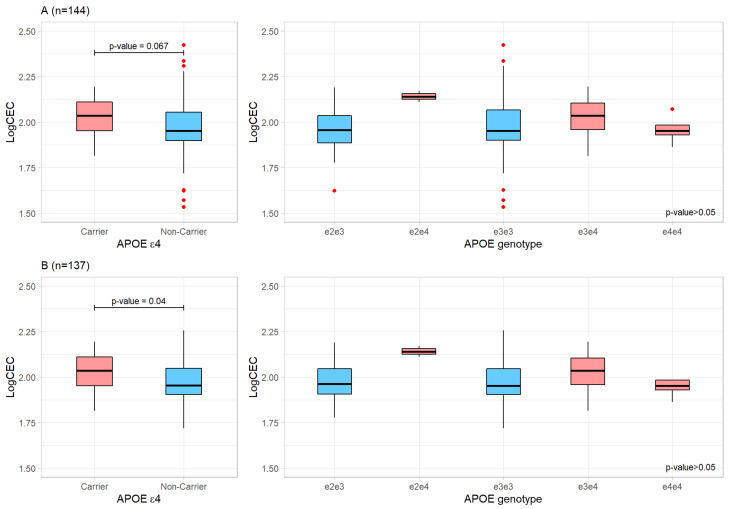
Cholesterol efflux capacity by *APOE* genotypes with (**A**) or without (**B**) CEC outliers. Left panels represent CEC values from *APOE* ε4 carriers vs. CEC values from *APOE* ε4 non-carriers. Right panels represent CEC levels in different *APOE* genotypes. Blue denotes non-ε4 categories. Red dots denote outliers identified.

**Figure 2 ijms-24-02186-f002:**
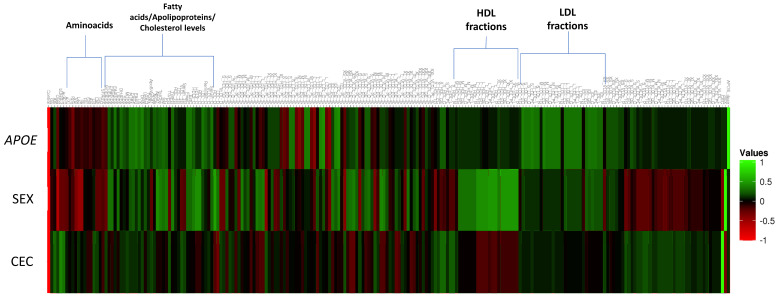
Global heatmap analyzing the patterns of correlations between plasma NMR features and *APOE*, SEX, and CEC. The details of the NMR notation are described in Appendix A.

**Figure 3 ijms-24-02186-f003:**
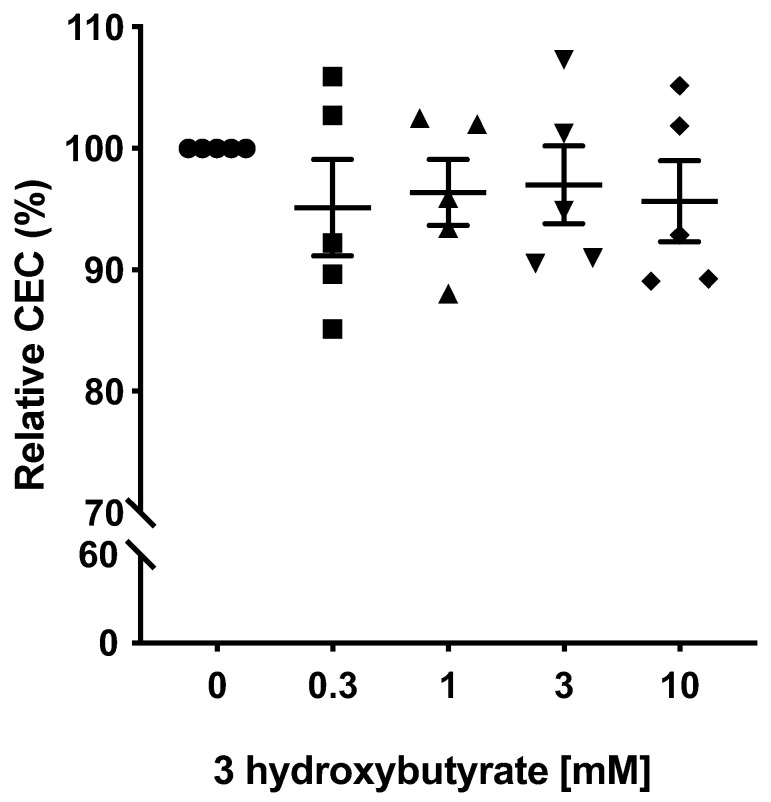
Effect of the supplementation of the Apo B-depleted plasma with (±)-sodium 3-hydorxybutyrate at different concentrations in CEC. Data were not statistically different (*p* = 0.84; Mann–Whitney test).

**Figure 4 ijms-24-02186-f004:**
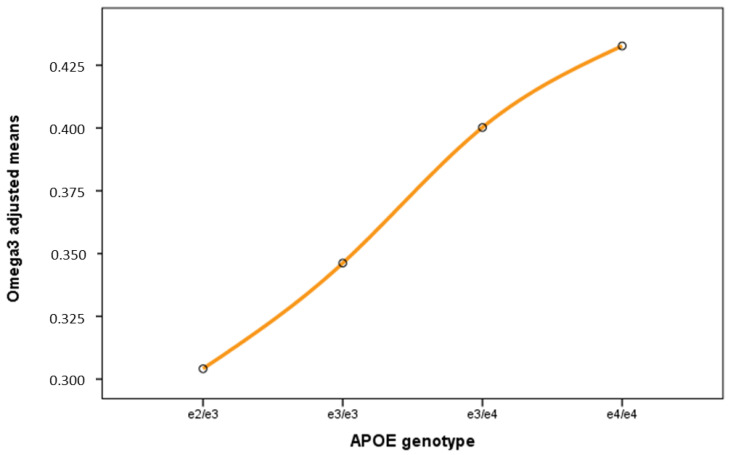
Adjusted effect of the *APOE* genotypes in the plasma levels of omega3 fatty acids (Nightingale FAw3). Sample size (n) 136. Adjusting covariates in the model are age, Sex, L-HDL-C(%), and personal antecedents of hypertension and dyslipidemia.

**Table 1 ijms-24-02186-t001:** Top variables associated to the cholesterol efflux capacity (CEC) in the studied population. The influence of the *APOE* genotype in the variables is also quantified.

Variable	Effect in CEC (beta)	*p*-Value (CEC Effect)	*APOE* Impact (beta)	*p*-Value (APOE Effect)
3-Hydroxybutyrate	0.365	0.0000073 *	0.025	0.765
Sex	−0.326	0.000068 *	0.096	0.254
Acetoacetate	0.317	0.00012 *	−0.013	0.88
Mean diameter for LDL particles	−0.282	0.001	−0.147	0.08
Albumin	0.279	0.001	0.147	0.08
Total cholesterol in small HDL particles	0.254	0.002	0.221	0.008
Cholesterol esters in small HDL particles	0.245	0.003	0.23	0.006
Concentration of small HDL particles	0.229	0.006	0.109	0.195
Total lipids in small HDL particles	0.229	0.006	0.117	0.165
Familial antecedents of Parkinson’s disease	−0.22	0.009	−0.18	0.034
Total-cholesterol-to-total-lipid ratio in very small VLDL particles	−0.216	0.009	0.156	0.064

* Indicates effects with statistical significance after Bonferroni correction.

**Table 2 ijms-24-02186-t002:** NMR features associated with the *APOE* locus and their effect on the CEC.

NMR Parameter	APOE (beta)	*p*-Value APOE	CEC (beta)	*p*-Value CEC
Total-cholesterol-to-total-lipid ratio in large LDL particles	0.406	0.00000054	0.002	0.984
Total-cholesterol-to-total-lipid ratio in IDL particles	0.405	0.00000058	−0.073	0.386
Triglyceride-to-total-lipid ratio in large LDL particles	−0.398	0.00000094	0.024	0.778
Triglyceride-to-total-lipid ratio in medium LDL particles	−0.386	0.000002	−0.008	0.925
Triglyceride-to-total-lipid ratio in IDL particles	−0.378	0.000004	0.094	0.262
Cholesterol-ester-to-total-lipid ratio in large LDL particles	0.376	0.000004	0.074	0.377
Cholesterol-ester-to-total-lipid ratio in IDL particles	0.375	0.000004	−0.003	0.968
FAw3: Omega-3 fatty acids	0.359	0.000011	0.105	0.212
Total-cholesterol-to-total-lipid ratio in medium LDL particles	0.357	0.000013	0.024	0.774
Free cholesterol in small LDL particles	0.346	0.000025	0.054	0.524
Cholesterol-ester-to-total-lipid ratio in medium LDL particles	0.337	0.000041	0.053	0.529
DHA: 22:6, docosahexaenoic acid	0.334	0.000048	0.123	0.142
Phospholipids in small LDL particles	0.334	0.00005	0.098	0.246
Free cholesterol in medium LDL particles	0.332	0.000055	0.055	0.516
EstC: Esterified cholesterol	0.32	0.000105	0	0.999
Total cholesterol in small LDL particles	0.319	0.000111	0.049	0.564
FAw3/FA: Ratio of omega-3 fatty acids to total fatty acids	0.318	0.000115	0.075	0.373
Total lipids in small LDL particles	0.317	0.000121	0.066	0.434
PUFAs: Polyunsaturated fatty acids	0.316	0.000129	0.09	0.284
Total cholesterol in medium LDL particles	0.316	0.000131	0.051	0.548
Serum total cholesterol	0.313	0.000148	0.001	0.994
Concentration of small LDL particles	0.312	0.000156	0.07	0.408
Total cholesterol in LDL particles	0.312	0.000157	0.033	0.698
Cholesterol esters in medium LDL particles	0.311	0.000166	0.049	0.557
Cholesterol esters in small LDL particles	0.311	0.000169	0.047	0.574
Phospholipids in medium LDL particles	0.31	0.000173	0.085	0.315
Total-cholesterol-to-total-lipid ratio in small LDL particles	0.308	0.000196	0	1
Phospholipids in large LDL particles	0.307	0.000203	0.026	0.757
Cholesterol esters in large LDL particles	0.306	0.000216	0.025	0.766
Total lipids in medium LDL particles	0.306	0.000218	0.06	0.48
Total cholesterol in large LDL particles	0.305	0.000219	0.016	0.852

A list of plasmatic NMR parameters with Bonferroni-corrected significant associations with the *APOE* genotype (*p* < 0.00022 for 246 comparisons). In contrast, none of them have a statistically significant effect on the CEC.

**Table 3 ijms-24-02186-t003:** Adjusted backward regression analysis of plasma NMR features, clinical, and demographics associated with the *APOE* dosage.

Variable	beta (Effect)	*p*-Value (beta)
(Regression constant)		0.697
Hypertension	−0.184	0.016
Dyslipidemia	0.137	0.091
Total-cholesterol-to-total-lipid ratio in IDL particles (IDL-C-%)	5.373	0.000016 *
Triglyceride-to-total-lipid ratio in large LDL particles (L-LDL-TG-%)	0.627	0.08
Cholesterol-ester-to-total-lipid ratio in large LDL particles (L-LDL-CE-%)	−4.202	0.02
Cholesterol-ester-to-total-lipid ratio in IDL particles (IDL-CE-%)	−2.395	0.003
Omega-3 fatty acids (FAw3)	1.718	0.002
Free cholesterol in small LDL particles (S-LDL-FC)	2.076	0.033
Cholesterol-ester-to-total-lipid ratio in medium LDL particles (M-LDL-CE-%)	8.641	0.001
Esterified cholesterol (EstC)	−4.59	0.000017 *
Ratio of omega-3 fatty acids to total fatty acids (FAw3/FA)	−0.944	0.024
Concentration of small LDL particles (S-LDL-P)	−5.025	0.091
Cholesterol esters in medium LDL particles (M-LDL-CE)	3.588	0.033
Phospholipids in medium LDL particles (M-LDL-PL)	3.228	0.012
Total-cholesterol-to-total-lipid ratio in small LDL particles (S-LDL-C-%)	−7.003	0.00018 *

* Statistical significance after Bonferroni correction (*p* < 0.00024). Negative effects indicate inverse association with e4 carrier status and vice versa.

**Table 4 ijms-24-02186-t004:** Demographics of the selected population.

Demographics	Valid Sample	Mean|*n*	Std|%
Age (years)	144	65.6	8.9
Sex (female)	144	82	56.9
Height (cm)	140	165.7	9.2
Weight (kg)	141	73.4	13.2
BMI (kg/m^2^)	140	26.6	3.97
Familiar antecedent dementia (% subjects)	137	90	65.7
Familial antecedent Parkinson’s disease (% subjects)	138	13	9.4
Dyslipidemia (% subjects)	137	64	46.7
Hypertension (% subjects)	138	46	33.7
*APOE* (E4 carriers) (% subjects)	143	31	21.7

## Data Availability

The datasets used and/or analyzed during the current study are available from the corresponding author on reasonable request.

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
