# Peer review of "Correlations between the NMR Lipoprotein Profile, APOE Genotype, and Cholesterol Efflux Capacity of Fasting Plasma from Cognitively Healthy Elderly Adults"

_ijms, 2023, doi:10.3390/ijms24032186_

Round 1

Reviewer 1 Report

Authors  aim to demonstrate that  poorely defined human serum components may affect cholesterol backward uptake from the cells. Rationale for choosing mice line J774 macrophages is justified by one, although important factor (ie lack apo-E expression). However,  existng multiple inter-species differences still may disturb multiparametric analysis. Thus, this inconsistency should  be carefully dicussed on the basis of existing reference data.  

Also  BHB concentration as  a strogest factor correlating, non causally, with CEC needs more careful elaboration.  We know no range of BHB concetrations in tested population. Therefore, in contrast to other correlations placed in the tables 2/3, it would need at least figure with CEC vs BHB correlation plot. If some HBH levels exceed known reference limits, they should be individually evaluated. or treated as outlayers or pathology.

Addition of very high BHB concentrations (Fig 3) markedly increases osmolality which in turn decreases J774 cell volume, what may artificially increase CEC. Methods contain no information on adjustment mediums osmolality. Much lower 20 mM BHB cocencentration is noncompatible with life and is not applicable even to cellular models. 

These points need to be rectified before re-considering paper for publication

Author Response

Authors  aim to demonstrate that  poorely defined human serum components may affect cholesterol backward uptake from the cells. Rationale for choosing mice line J774 macrophages is justified by one, although important factor (ie lack apo-E expression). However,  existng multiple inter-species differences still may disturb multiparametric analysis. Thus, this inconsistency should  be carefully dicussed on the basis of existing reference data. 

Answer: we thank to reviewer for his/her comments on our manuscript. The use of this model for cholesterol efflux is considered standard in the field. In fact we implemented the same procedure previously used which has demonstrated important endophenotypes related to cardiovascular disorders (see N Engl J Med 2011; 364:127-135) . Most important, the technique is solely implemented to check differential acceptor capacity of human sera by using a common, robust and standard cell model. So, the scope of our study is not investigating the J774 macrophages CEC model itself but identifying properties or components in the acceptor sera that might be influencing this very standard experiment.  So, although we are aware of the importance of the cell model for this type of studies, we consider the proposed discussion beyond the scope of our work.

Also  BHB concentration as  a strogest factor correlating, non causally, with CEC needs more careful elaboration.  We know no range of BHB concetrations in tested population. Therefore, in contrast to other correlations placed in the tables 2/3, it would need at least figure with CEC vs BHB correlation plot. If some HBH levels exceed known reference limits, they should be individually evaluated. or treated as outlayers or pathology.

Answer: thanks for this commentary. We have added the range of BHB concentrations to the text, plotted the CEC vs BHBs correlation in the supplementary information and conducted sensitivity analyses removing sequentially BHB concentration outliers over 4SD, 3SD, 2SD and 1SD BHB values. Our results, even with the most pruned calculation (1SD), demonstrates statistically significant correlation between BHB levels and CEC. So, our observation is not depending solely on extreme BHB values.  Evidently, the sequential removal of extreme values progressively reduces the effect size estimate. We feel that these sensitivity analyses reinforce the connection between CEC and BHB levels. 

Phenotype CEC

All individuals

4 SDV removed

3 SDV removed

2 SDV removed

1 SDV removed

R Pearson Correlation

0.365

0.289

0.281

0.258

0.231

P-value

<0.001

0.001

0.001

0.003

0.008

Sample size (N)

143

140

137

134

130

Addition of very high BHB concentrations (Fig 3) markedly increases osmolality which in turn decreases J774 cell volume, what may artificially increase CEC. Methods contain no information on adjustment mediums osmolality. Much lower 20 mM BHB cocencentration is noncompatible with life and is not applicable even to cellular models.

We thanks the reviewer for calling our attention to this important point. We are aware that BHB levels >25 mM are not usually found in human blood, even in pathological conditions (diabetic ketoacidosis). However, we want to see whether BHB could affect CEC even at very high concentrations. We nonetheless concurs that this could be misleading, and therefore have changed the graph in Fig. 3 to show the results in the range of BHB concentrations compatible with other reports showing BHB effects in vitro.

Para incluir en la fig. legend. Data were not statistically different (p = 0.84; Mann-Whitney test).

These points need to be rectified before re-considering paper for publication

Reviewer 2 Report

General comment

This paper is dealing with the effects of lipoproteins, metabolites and genetic parameters on cholesterol ester efflux capacity; this is a highly relevant issue which surely deserves investigation; however, a relatively low number of patients together with obvious technical difficulties are questioning the reported experimental results and conclusions drawn from these data.

Specific points

1) A significant part of cholesterol efflux capacity is mediated by scavenger receptor BI (SR-BI), which has been totally neglected within this paper.

2) A large GWAS and GWES study already showed the association of CEC and ApoE gentoypes in more than 5000 patients (Low-Cam et al, JAHA 2018, DOI 10.1161). This paper is not included as reference. The statement that APOE genotype has only a weak effect of CEC appears to be at least misleading on this basis.

3) There are obviously major difficulties concerning the CEC technology used. As this is a central point of this work the whole meaning of results and conclusion drawn has to be questioned:

- It is not described how BODIPY labeled LDL was generated and how much of this labeled LDL has been used for the assay.

- What do the authors mean with “washing the reaction mixture with serum-free medium”? This does not make any sense.

- Authors describe that “The fluorescence of media containing BODIPY-cholesterol was subtracted from that of media with no BODIPY-cholesterol”. This again makes no sense.

- What are  “time-zero monolayer fluorescence values before and after cell lysis”? Usually the CEC is calculated as the fluorescence of the acceptor containing supernatant divided by the  sum of this parameter with fluorescence of the remaining cell extract.

- Why is there a need for normalization to a pooled reference apolipoprotein depleted plasma?

- The intraplate coefficient of variation is very high (33.37 !!!); this does not allow collection of relevant data.

- How many outliers have been removed, what was the criteria for doing this.

4) Table 1: Dimensions of most variables are missing; numbers? Percentage?

Author Response

General comment

This paper is dealing with the effects of lipoproteins, metabolites and genetic parameters on cholesterol ester efflux capacity; this is a highly relevant issue which surely deserves investigation; however, a relatively low number of patients together with obvious technical difficulties are questioning the reported experimental results and conclusions drawn from these data.

Answer:  Thanks for this general comment. Limitations section has been enhanced to explain the difficulties inherent to this study including sample size and experimental approaches.

Specific points

1) A significant part of cholesterol efflux capacity is mediated by scavenger receptor BI (SR-BI), which has been totally neglected within this paper.

Asnwer: Thanks for highlighting this interesting mechanism.  We added it to introduction section. However, as mentioned in the reviewer1 answer, the molecular dissection of the CEC model is beyond the scope of this work. We just focus on the potential characteristics of the acceptor sample to modify the CEC in a homogeneous and established model.

2) A large GWAS and GWES study already showed the association of CEC and ApoE gentoypes in more than 5000 patients (Low-Cam et al, JAHA 2018, DOI 10.1161). This paper is not included as reference. The statement that APOE genotype has only a weak effect of CEC appears to be at least misleading on this basis.

 Answer: thanks for advising on this interesting JAHA manuscript.  Please notice that this is reference in number 20 of our manuscript in the text. Please notice also that reported beta coefficient for CEC variants associated to J774 stimulated CEC is just -0.193 which represents a small effect (less than 0.2SDV of CEC per rs445925 allele). Please also notice that this variant is not associated to Alzheimer’s disease (see figure 1 extracted from gwas catalog). Indeed, rs445925 has low LD with APOE AD related SNPS (r2<0.6, figure 1). In contrast, this variant is highly associated to a number of lipid metabolites in different studies (see https://www.ebi.ac.uk/gwas/variants/rs445925 for further details).

Figure 1: Regional plot obtained from GWAS catalog. The figure is indicating the degree of Linkage disequilibrium (LD between rs445925 and Alzheimer disease SNPs represented as Orange dots. No AD SNP is proxy of the top variant associated with CEC by Low-Kam et alia.

In sum, all these observations reinforce the notion that reported loci are linked to CEC via the change in the lipid profile of the acceptor. The lack of association to AD of the key SNP observed is also making even more difficult the detection of the APOE locus effect on CEC using AD-related variants. So, we feel we are not powered enough to reveal such effects using both our current sample size and the selected SNPs (comprising APOE AD-related haplotypes). We have made the appropriate amendments in the text to reassure these limitations.

3) There are obviously major difficulties concerning the CEC technology used. As this is a central point of this work the whole meaning of results and conclusion drawn has to be questioned:

- It is not described how BODIPY labeled LDL was generated and how much of this labeled LDL has been used for the assay.

Answer: We apologize for this mistake, but it should read BODIPY-labeled cholesterol. This has been corrected in the text

- What do the authors mean with “washing the reaction mixture with serum-free medium”? This does not make any sense.

Answer: Our apologizes again for this mistake. This protocol has been profoundly rewrote to permit the correct interpretation of the assay.

- Authors describe that “The fluorescence of media containing BODIPY-cholesterol was subtracted from that of media with no BODIPY-cholesterol”. This again makes no sense.- What are  “time-zero monolayer fluorescence values before and after cell lysis”? Usually the CEC is calculated as the fluorescence of the acceptor containing supernatant divided by the  sum of this parameter with fluorescence of the remaining cell extract.

Answer: The section on CEC determination in Methods has been rewrote to clarify all these points. We hope now it is clearer .

- Why is there a need for normalization to a pooled reference apolipoprotein depleted plasma?

Answer:To correct for the inter-assay variability since it is not possible to run all plasma samples in the same assay. This point has been explained now in Methods.

- The intraplate coefficient of variation is very high (33.37 !!!); this does not allow collection of relevant data.

Answer: We apologize because we changed inter- and intra-assay values; this has been corrected in the new version of the manuscript. In addition, we have recalculated the inter-assay CV using only CEC values for the pooled plasma. The value obtained is now 18.9%, which has been included in the same section.

- How many outliers have been removed, what was the criteria for doing this.

Answer: Outliers were identified separately in APOE4 carriers and non-carriers groups using the R base function `boxplot.stats` with standard parameters (coef=1.5). Then, 7 individuals from the APOE4 non-carriers where removed from further analysis. We have added this detail in methods.

4) Table 1: Dimensions of most variables are missing; numbers? Percentage?

Answer: Thanks for advising us. We have added the dimensions of all variables In table 1.

Round 2

Reviewer 1 Report

Improvements have met  reviewer's rmerks.

Author Response

Thanks

Reviewer 2 Report

The manuscript has been improved; however, two major points were not answered adequately:

1) SR-BI may not considered as an alternative efflux mechanism but rather has been established as a major player within the reverse cholesterol transport; this should at least adequately be considered within the Introduction and The Discussion.

 2) Removal of outliers regarding the analysis of ApoE genotype and CEC should be documented within the Results; I recommend presentation of the analysis with and without these outliers in parallel (also graphically).

Author Response

1) SR-BI may not considered as an alternative efflux mechanism but rather has been established as a major player within the reverse cholesterol transport; this should at least adequately be considered within the Introduction and The Discussion.

Answer: The introduction and discussion sections have been modified to reinforce the role of SR-BI in CEC.

 2) Removal of outliers regarding the analysis of ApoE genotype and CEC should be documented within the Results; I recommend presentation of the analysis with and without these outliers in parallel (also graphically).

Answer: Figure 1 has been amended to present the analyses with and without outliers. 

Round 3

Reviewer 2 Report

1) The introduction has been improved.

2) Significance levels should be given for both models in Figure 1.

3) Page 5, first line: authors state "values for noncarriers were higher"; the mean value for noncarriers appear to be lower, please explain.

Author Response

1) The introduction has been improved.
Answer: Thanks to the reviewer for advicing us on this issue.
2) Significance levels should be given for both models in Figure 1.
Answer: We have amended the figure 1 and provided p-values for each model
3) Page 5, first line: authors state "values for noncarriers were higher"; the mean value for noncarriers appear to be lower, please explain.
Answer: This just a mistake. We have aligned the meaning of the sentence to the observed results in the text.